# Seroprevalence of antibodies against SARS-CoV-2 virus in Northern Serbia (Vojvodina): A four consecutive sentinel population-based survey study

Mioljub Ristić[1,2]*, Biljana Milosavljević[2], Slobodanka Vapa[2], Miloš Marković[3], Vladimir Petrović[1,2]

1 Department of Epidemiology, Faculty of Medicine, University of Novi Sad, Novi Sad, Serbia, 2 Institute of Public Health of Vojvodina, Novi Sad, Serbia, 3 Department of Immunology, Faculty of Medicine, Institute of Microbiology and Immunology, University of Belgrade, Belgrade, Serbia

* mioljub.ristic@mf.uns.ac.rs

## Abstract

### Background

Monitoring changes of seroprevalence over time is important at the beginning and during of COVID-19 outbreak to anticipate its dynamics and plan an adequate public health response.

### Methods

We conducted a repeated cross-sectional investigation among asymptomatic outpatient subjects and covered 0.1% of total population of Northern Serbia (Autonomous Province of Vojvodina). Each participant was tested for anti-SARS-CoV-2 antibodies using an immuno-chromatographic qualitative test (point-of-care rapid test). In the last round of survey IgG antibodies targeting the S1 subunit of the spike protein and the nucleocapsid protein of SARS-CoV-2 virus were assessed.

### Results

During the four rounds of survey (between the end of April and the end of September), anti-SARS-CoV-2 seropositivities based on immunochromatographic test results were 2.60% (95% CI 1.80–3.63), 3.93% (95% CI 2.85–5.28), 6.11% (95% CI 4.72–7.77) and 14.60% (95% 12.51–16.89), respectively. After adjusting with results obtained from the Line immunoassay test, the estimated overall seroprevalence increased to 16.67% (95% CI 14.45–19.08) corresponding to 322,033 infections in total by the end of September 2020 in Vojvodina's population. Throughout the course of the study, for every RT-PCR confirmed case of COVID-19, there were 39–87 additional infections in Vojvodina. No significant difference (p>0.05) in SARS-CoV-2 seropositivity regarding gender and between age subgroups was observed over the course of the survey.

**Data Availability Statement:** All relevant data are within the manuscript and its Supporting information files.

**Funding:** The funders had no role in study design, data collection and analysis, decision to publish, or preparation of the manuscript. The authors received no specific funding for this work.

**Competing interests:** The authors have declared that no competing interests exist.

## Conclusions

The population prevalence of SARS-CoV-2 antibodies implies much more widespread infection in Vojvodina than indicated by the number of confirmed cases. However, our results suggest that the population of Vojvodina has not reached a desirable level of protection from SARS-CoV-2 virus by the end of September 2020.

## Introduction

Public health surveillance of coronavirus disease 2019 (COVID-19) in humans caused by infection with severe acute respiratory syndrome coronavirus 2 (SARS-CoV-2) both on global and local levels is of great importance for mitigation and cessation of the ongoing pandemic. So far, initial surveillance of SARS-CoV-2 infections across globe has focused primarily on patients with severe disease or case-based surveillance, while mildly affected, asymptomatic individuals and those who have not been tested due to limited availability of tests generally have not been screened [1–3].

According to current opinion [1, 4], there is no reason to wait for the end of the epidemics before doing seroprevalence studies across the world. Moreover, improved serological survey of specific antibodies against SARS-CoV-2 virus has been proposed with the aim to estimate the cumulative prevalence of asymptomatic infection in the community, and to better clarify the dynamics of the epidemics. Although it is still not clear whether antibodies against SARS-CoV-2 correlate with protective immunity, survey study of SARS-CoV-2 antibodies among residents in certain territory can offer the possibility to calculate approximately the number of those who could potentially exhibit immunologic protection against subsequent infection [1, 5–8]. The available serological tests for detection of antibodies (IgM or IgG) against SARS-CoV-2 are helpful to provide a more precise estimate of the cumulative number of subjects who have been in contact with SARS-CoV-2 virus compared to those detected by molecular diagnostic tests [6].

As recommended by World Health Organization (WHO), implementation of the unique protocol with the purpose to investigate the extent of infection is desirable in every country in which SARS-CoV-2 virus infection has been reported [1]. In addition, monitoring changes of seroprevalence over time is important at the beginning and during the COVID-19 outbreak to anticipate its dynamics and plan the adequate public health response [1, 2].

In the Northern Serbia (Autonomous Province of Vojvodina with total population of 1.9 million people), the first (imported) laboratory-confirmed case of COVID-19 was registered on March 6, 2020. At the same time, this was the first confirmed case of SAR-CoV-2 infection in Serbia Several days later, on March 15, 2020, a lockdown was implemented in the whole territory of Serbia, which included closing of the borders, a strict movement restrictions (social distancing for all citizens), case isolation with home quarantine, as well as closure of schools, kindergartens and faculties. After May 6, 2020, there was a relaxation of epidemiological measures (only case isolation and household quarantine remained) with prevention of the spread of the SARS-CoV-2 infection based only on recommendations leading to the second epidemic wave that lasted between mid-June and until mid-September. Vaccination against COVID-19 in Serbia started on December 24, 2020, almost three months after the end of this study [9]. Throughout the COVID-19 outbreak in Vojvodina, a RT-PCR test has been permanently available for laboratory confirmation of infection. An immunochromatographic qualitative assay for detection of both IgG and IgM anti-SARS-CoV-2 antibodies was also used in parallel

with RT-PCR [9, 10]. As a result of implementation of different testing criteria in our country during the course of epidemic, it is likely that many asymptomatic and mild SARS-CoV-2 infections were underreported, similar to situations observed in other regions [1–7]. On the other hand, appropriate insight in the real extent of SARS-CoV-2 infection is critical for an effective public health response to COVID-19 in the upcoming epidemic waves [1–7].

Even though the epidemic of COVID-19 in Serbia started in the beginning of March [9], no seroprevalence data are available yet. The aim of this study was to assess the dynamics of SARS-CoV-2 antibodies seroprevalence over time through repeated cross-sectional surveys and to estimate the extent of infection among individuals without previous history of COVID-19 diagnosis in Vojvodina. In addition, this study aimed to examine the usefulness of sentinel surveillance of acute respiratory infections based on comparison of estimated cases detected through this surveillance system and those noticed through serosurvey.

## Materials and methods

### Study design and participants

This is a prospective study performed according to WHO protocol as repeated cross-sectional investigation in the same geographic area [1]. Data obtained in this manner were used to estimate the prevalence of SARS-CoV-2 antibodies in general population (non-hospitalized persons) of Vojvodina. To this end, the Institute of Public Health of Vojvodina, Novi Sad in collaboration with six local departments of public health of Vojvodina coordinated a training of health professionals in those local departments of public health as well as at 44 Government Primary Health Care Centres that are included in Vojvodina Health Care System. The sample size was predefined and 135 sentinel physicians were involved at primary (outpatient) medical care sites, altogether covering 0.1% out of the total population (1,232,867) from 44 settlement areas in Vojvodina. The same experienced physicians had been previously included in the active surveillance of influenza-like illness (ILI) and acute respiratory infection (ARI) in Vojvodina, as a part of influenza surveillance system that was described in detail in previous studies [11, 12]. Briefly, the surveillance system used the network of sentinel physicians (general practitioners and paediatricians) who recruited the representative sample of participants on a voluntary basis after controlling for age and gender. A stratified random sample from each of 44 settlement areas was involved, with enrolment for subgroups based on age (0–4, 5–14, 15–29, 30–65 and ≥65 years), and gender distribution of Vojvodina.

Only respondents without any COVID-19-related symptoms within 30 days and without known direct contact with laboratory-confirmed COVID-19 case in the past 14 days were invited to participate in the study by their sentinel physicians. In addition, only two healthy participants per family were included. Among participants, there were no health care workers. During the COVID-19 lockdown period for elderly persons in Vojvodina, health care personnel collected samples from this population at their homes. After relaxation of epidemic control measures, they were recruited at primary care level settings, as others. All participants were provided with a surgical mask before the entrance to the sentinel site. During the visit, sentinel physicians explained the aims and purpose of the study once again to potential participants to ensure their informed consent. Upon sample collection, at least two previously trained medical staff practitioners analysed each serological test, and the participants or their parents or guardians were promptly informed about the test results thereafter.

Following the recommendation of WHO that cross-sectional studies can be completed after the peak of transmission of the epidemic wave, but also at any time during the epidemic [1], our study was performed in four successive follow-up rounds (at the end of April, May, June and September). Taking into account known facts that a vast majority of tested patients

with COVID-19 (in some studies 100%) developed IgG antibodies against the SARS-CoV-2 within three weeks after symptom onset with median day of seroconversion for both IgG and IgM being 13 days after the onset of symptoms [13], we have estimated the seroprevalence in accordance with the epidemiological situation in Vojvodina, i.e. during initial epidemic period (I round), after the first wave (II round), and before and after the second wave (III and IV rounds) of COVID-19 epidemic.

Each round was scheduled to be completed within one week. At the beginning of the study, a total of 1,267 participants from 44 settlements in Vojvodina have been included, representing at least 0.1% of the total population of the province. During the next rounds of survey, we excluded participants who had IgM or/and IgG antibodies against SARS-CoV-2 previously confirmed. The efforts were undertaken to replace omitted subjects with other respondents matched by age, gender and settlement area in Vojvodina, but despite all the efforts, during the last round of survey, only 1,041 subjects were involved.

**Laboratory testing.** Blood samples were obtained from each participant aseptically using either a finger prick (for children) or venipuncture technique (for adults) in local Primary Health Care sentinel sites. The presence of anti-SARS-CoV-2 IgM or/and IgG antibodies were assessed using commercially available immunochromatographic qualitative assay (point-of-care rapid test). In the fourth round of survey, anti-SARS-CoV-2 IgG antibodies were further confirmed by Line immunoassay targeting various proteins of SARS-CoV-2.

*Immunochromatographic qualitative test.* A rapid lateral one-step flow chromatographic immunoassay using colloidal gold-labeled SARS-CoV-2 antigens (Innovita 2019-nCoV Ab Test, Innovita Biological Technology Co., Ltd., China) was used in this study following manufacturer's recommendations. This point-of-care rapid test allows rapid (in 10 minutes) qualitative detection and differentiation of IgM and IgG of antibodies against SARS-CoV-2 virus. The test line was precoated with anti-human IgM + IgG antibodies. No lines in the result window are visible prior to applying any specimen. In the case that sample is positive for IgM or IgG antibodies against SARS-CoV-2, the "colloidal gold-conjugated SARS-CoV-2 antigen–anti-SARS-CoV-2 antibody" complex is bound by capture antibodies coated on the test line and a burgundy-coloured band is developed. If SARS-CoV-2 antibodies are not present in the specimen, no colour will appear in the test line [14]. This test was approved by the Chinese Food and Drug Administration and by US FDA. The reported sensitivity and specificity of this immunoassay test are 87.3% and 100%, respectively [15].

*Line immunoassay quantitative test.* Detection and avidity determination of SARS-CoV-2 IgG antibodies were further assessed using a commercially available recomLine SARS-CoV-2 IgG [Aviditat] (Mikrogen Gmbh; Germany) according to manufacturer's recommendations. The test allows identification of specific antibodies against the individual antigens for SARS-CoV-2, namely nucleocapsid protein (NP), receptor binding domain of the spike protein (RBD) and S1 subunit of the spike protein (S1). Antibodies against NP antigens of seasonal human coronaviruses (HCoV: NL63, OC43, 229E, and HKU1) are also detected.

The test was performed using a DYNABLOT Plus test (Dynex, Czech Republic) by incubation of two test strips with diluted serum sample (1:100 dilution). During the first incubation, antibodies in the sample bind to their specific antigens that are fixed on the test strips. Then one of the two test strips is washed with the avidity solution. During this step, the low avidity antibodies diffuse away, while high avidity antibodies remain bound to their specific antibodies. After washing, anti-human antibodies conjugated with horseradish peroxidase are added. The specific antibodies bound to their antigens are visualised after repeated round of washing with the addition of a substrate that makes the antigen-antibody complexes visible as bands that appear on the test strips. The relative position of the stained bands indicates the specificity of the reacting antibodies.

The test strips were analysed using the *recomScan* test strip analysis software version 3.4.162 and the intensities of the corresponding bands on the two test strips (IgG strip and avidity strip) incubated with the same patient sample were compared and the changes noted. By subsequently comparing the two corresponding test strips, the avidity of the antibodies was determined and the particular avidity index was calculated. Reduction in the intensity of the SARS-CoV-2 bands (corresponding to NP, RBD and S1 antigens) or the HCoV bands (NP antigen) by more than 60% (avidity index ≤ 0.4) was considered as low avidity IgG antibodies, while reduction between 40% and 60% was considered intermediate (avidity index between 0.4 and 0.6). The reduction of the band intensity of less than 40% (avidity index ≥ 0.6) was considered as high avidity anti-SARS-CoV-2 antibodies [16].

## Data analysis

Similar to previously used methodology [11, 12], the population under sentinel surveillance at primary care level in Vojvodina was used as a denominator for calculations of the weekly incidence of ILI and ARI per 100,000 inhabitants, while a numerator was the number of clinical cases of ILI and ARI in the total population from March 6 till the end of October, 2020. The expected number of patients with ARI and COVID-19 among inhabitants of Vojvodina was extrapolated from the results obtained during the sentinel surveillance of ARI and anti-SARS-CoV-2 seropositivity rates throughout this survey. Cumulative number of ARI cases and estimated number of subjects with previous contact with SARS-CoV-2 virus obtained by serosurvey were compared (as ratio) with the cumulative number of officially registered laboratory-confirmed COVID-19 cases.

The proportions of positive tests (either IgM or IgG) given by point-of-care test in the analysis sample were calculated. In the last round of the serosurvey, these results were further adjusted according to the results provided by the more specific line immunoassay quantitative test, designed to detect high-avidity antibodies against various SARS-CoV-2 antigens and distinguish them from possibly cross-reactive antibodies to seasonal human coronaviruses. Tests of proportion were performed to compare values of seroprevalence by age group and gender of participants. As reference age and gender groups, we used the largest sample size. The stratum seroprevalence and 95% confidence intervals (CIs) of SARS-CoV-2 seropositivity were calculated using the SPSS software tool (version 22.0) MedCalc for Windows, version 12.3.0 (MedCalc Software, Mariakerke, Belgium). Statistical significance was set at $p < 0.05$.

## Ethical considerations

This investigation is considered a public health surveillance according to recommendation of WHO [1], and no clearance by Ethics Committee for this emergency response was required in Serbia. Before enrolment, oral informed consent from each participant or their parents or legal guardians (for participants under 15 years of age) was obtained. Personal and confidential information were removed, except for demographic information, including date of sampling, settlement area, age and gender of participants. No authors of this study treated the patients included in the analysis, and the data were anonymized before the authors accessed it.

## Results

The COVID-19 epidemic curve with the number of laboratory-confirmed cases plotted by date of patient onset of symptoms from March 6 (first reported case of COVID-19 in Serbia) to October 31, 2020 is shown in Fig 1. During this period, a total of 9,734 laboratory-confirmed COVID-19 cases were recorded in Vojvodina. The peak onset of cases was recorded on July 23, 2020, with a total of 256 cases confirmed that day.

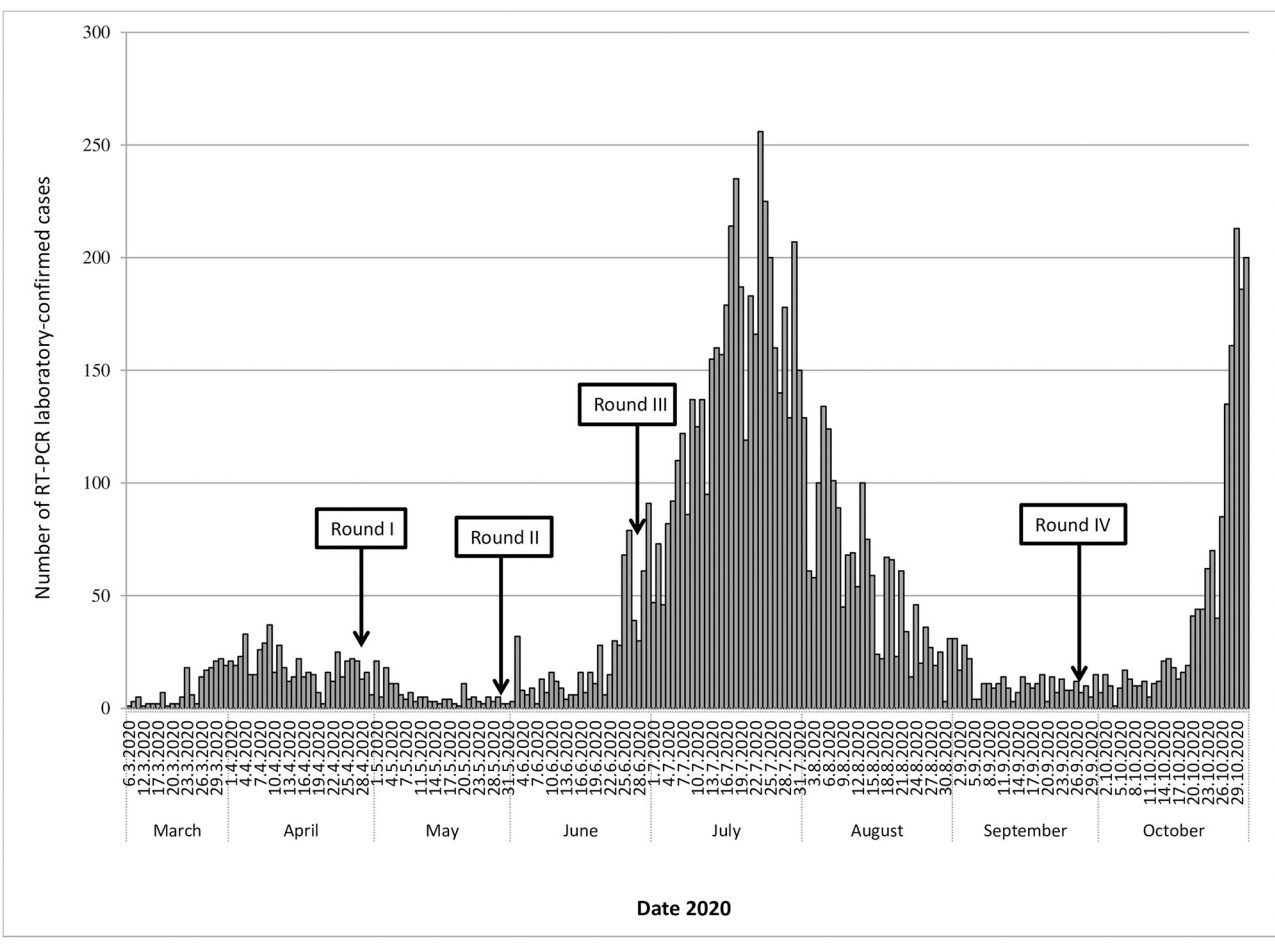

**Fig 1. Daily RT-PCR-confirmed COVID-19 cases reported and four time points of seroprevalence survey between March and October 2020 in Vojvodina, Serbia.**

There were three waves of COVID-19 disease in Serbia. The first wave lasted between March and the middle of May, while the second one lasted between mid-June and mid-September. The third wave started in the second half of October and is still ongoing. So far, the majority of confirmed cases in Vojvodina (7,204 of 9734 or 74.0%) were registered during the second wave. As indicated in Fig 1, four rounds of survey of SARS-CoV-2 antibodies were performed with the first one being done in April during the first wave, followed by the rounds at the end of the first wave, just before the second wave and after the second wave was over.

The levels of seroprevalence of SARS-CoV-2 antibodies in the population of Vojvodina are shown in Table 1. Over the four rounds of the serosurvey, between 1,014 (round III) and 1,267 (round I) participants were included. During the study period, between the end of April and the end of September, the levels of anti-SARS-CoV-2 seropositivity were 2.60% (95% CI 1.80–3.63), 3.93% (95% CI 2.85–5.28), 6.11% (95% CI 4.72–7.77) and 14.60% (95% CI 12.51–16.89), respectively. After adjusting for the results obtained with the Line immunoassay test (*spike protein* analysis), the estimated overall seroprevalence of Vojvodina's population at the end of September increased to the level of 16.67% (95% CI 14.45–19.08). Thus, a marked surge in seropositivity levels during the second wave was observed among the citizens of Vojvodina, since it was only 6.11% after the first wave of COVID-19 epidemic, but it has reached 16.67% once the second wave was over. Although the prevalence of SARS-CoV-2 seropositivity was

**Table 1. Overview of four consecutive seroprevalence surveys by age and gender between April and September, 2020 in Vojvodina, Serbia.**

| Characteristics | | Round I (End of April) [a] | | Round II (End of May) [a] | | Round III (End of June) [a] | | Round IV (End of September) [a] | | Adjustment results of seroprevalence (End of September) [b] |
|---|---|---|---|---|---|---|---|---|---|---|
| | | Tested | Seroprevalence % (95% CI) | Tested | Seroprevalence % (95% CI) | Tested | Seroprevalence % (95% CI) | Tested | Seroprevalence % (95% CI) | Seroprevalence % (95% CI) |
| Age group (years) | 0–4 | 52 | 0 | 34 | 0 | 32 | 0 | 29 | 10.34 (2.18–27.35) | 7.92 (1.20–24.16) |
| | 5–14 | 118 | 1.69 (0.20–5.98) | 71 | 2.82 (0.34–9.81) | 67 | 2.99 (0.37–10.38) | 77 | 12.99 (6.41–22.59) | 12.67 (6.18–22.21) |
| | 15–29 | 226 | 1.77 (0.48–4.47) | 177 | 3.39 (1.25–7.23) | 173 | 6.36 (3.22–11.09) | 173 | 16.76 (11.52–23.18) | 21.24 (15.40–28.09) |
| | 30–64 [c] | 679 | 3.39 (2.16–5.04) | 601 | 4.66 (3.12–6.66) | 563 | 6.93 (4.97–9.35) | 570 | 16.14 (13.21–19.42) | 17.34 (14.32–20.70) |
| | ≥65 | 192 | 3.13 (1.14–6.64) | 185 | 3.24 (1.20–6.92) | 179 | 5.59 (2.71–10.04) | 192 | 9.38 (5.66–14.41) [e] | 13.42 (8.94–19.07) |
| Gender | Male | 567 | 1.90 (0.94–3.39) | 480 | 2.48 (1.28–4.30) | 487 | 4.58 (2.90–6.83) | 458 | 13.48 (10.49–16.95) | 15.15 (11.99–18.77) |
| | Female [d] | 700 | 3.14 (1.98–4.72) | 588 | 4.09 (2.64–6.02) | 527 | 6.08 (4.20–8.47) | 583 | 15.49 (12.65–18.69) | 17.26 (14.28–20.58) |
| Overall | | 1267 | 2.60 (1.80–3.63) | 1068 | 3.93 (2.85–5.28) | 1014 | 6.11 (4.72–7.77) | 1041 | 14.60 (12.51–16.89) | 16.67 (14.45–19.08) |

[a] Immunochromatographic (point-of-care) qualitative test;

[b] Line immunoassay test;

[c] Reference age group;

[d] Reference gender group;

[e] Values that differ significantly compared with the 30–64 age group.

higher in females compared to males in all time points of the survey, those differences did not reach statistical significance (p>0.05). Among the youngest age group (0–4 years), the SARS-CoV-2 antibodies were not detected at all before the fourth circle of the survey. Interestingly, after adjustment of the results based on *spike protein* analysis, children aged 0–4 and 5–14 years of age had lower SARS-CoV-2 seropositivity (7.92%; 95% CI 1.20–24.16 and 12.67%; 95% CI 6.18–22.21, respectively) compared with those obtained by an immunochromatographic qualitative test (10.34%; 95% CI 2.18–27.35 and 12.99%; 95% CI 6.41–22.59, respectively). Based on immunochromatographic test results, the probability of being seropositive was significantly (p = 0.0212) lower in participants aged ≥65 years in comparison with those aged 30–64 years. However, the analysis of the age groups using the *spike protein* test failed to confirm this finding (p = 0.2043). Finally, there were no significant differences (p>0.05) in SARS-CoV-2 seropositivity between other age subgroups during the course of the survey.

In order to estimate the probable number of COVID-19 cases in Vojvodina, we analysed the results of the three types of surveillance systems (ILI, ARI and COVID-19 cases per 100,000 inhabitants) during the study period (Fig 2). According to these results, the trends of incidence for ILI and COVID-19 were apparently similar. As expected, compared to the incidence of COVID-19 in Vojvodina, the incidence rates of ARI were several times higher. These differences were particularly noticeable at the beginning as well as at the end of the observed period. Regarding the season of the year when ARIs are not common (between June and August), the incidence of ARI was six to 73 times higher than the incidence rates of COVID-19 per week in Vojvodina. Finally, the linear trend of difference between cases of ARI and COVID-19 cases registered in the surveillance system increased (y = 167.08x+6411.8, $R^2$ = 0.0805) over time (Fig 2).

We further compared the estimated cumulative numbers of COVID-19 and ARI cases, based on officially registered incidence rates, with the extrapolated number of SARS-CoV-2

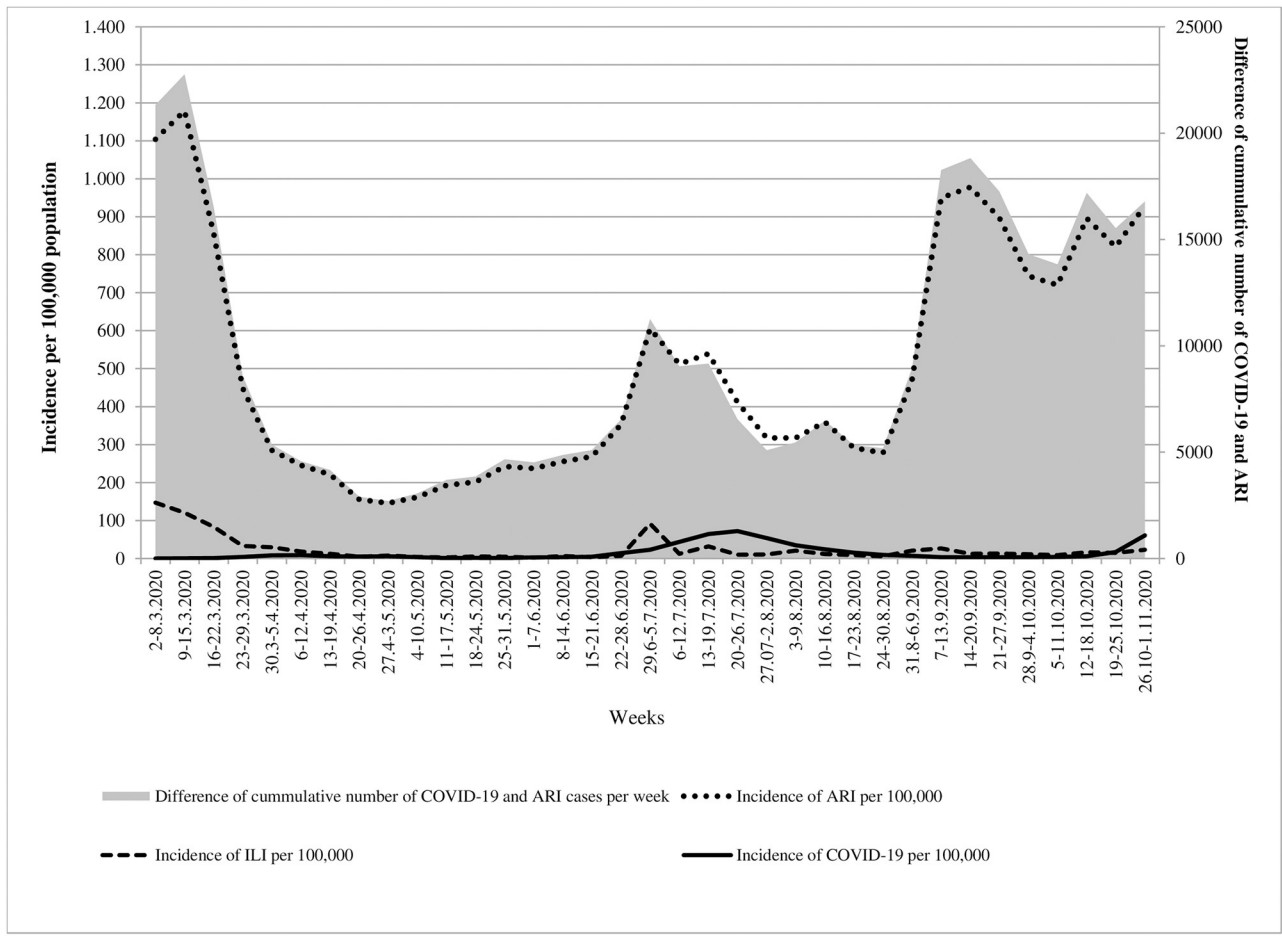

**Fig 2. Overview of the three different surveillance systems among patients with symptoms related to COVID-19 between March and October 2020 in Vojvodina, Serbia.** ARI: acute respiratory infection; ILI: influenza-like illness.

infections, as deduced from the results of seroprevalence survey. Cumulative and estimated COVID-19 and ARI cases during the study period are presented in Table 2. Overall, the differences between infections (based on anti-SARS-CoV-2 positivity) and COVID-19 cases (based on RT-PCR results) were higher during the second and third rounds (86.8 and 85.5, respectively) than during the first and the last survey cycles (75.9 and 35.5, respectively). As for the age groups, the highest differences of infection to case ratio throughout four seroprevalence rounds were recorded in the 5–14 age group (227.9, 221.8, 176.3 and 318.4, respectively). Concerning the ARI and case ratio, it was the highest in the children aged 0–4 years (1860.0, 1176.7 and 1039.1, respectively for the first three time points) with the exception for the last round of survey when it was the highest in 5–14 years age group (654.4). Also, the overall ARI to case ratios compared to infection to case ratios were higher in the first two rounds (119.1 vs 75.9, and 106.2 vs 86.8, respectively), but not in the last two rounds of the survey (79.2 vs. 85.5 and 25.5 vs. 34.5, respectively).

Considering the values of estimated number of subjects with SARS-CoV-2 antibodies during the last round of investigation adjusted with *spike protein* analysis, the differences in the estimated number declined in 0–4 and 5–14 age groups, but increased in other age groups compared to the values based on the immunochromatographic test results alone.

**Table 2. Cumulative and estimated number of COVID-19 and acute respiratory infection cases and their ratio between April and September, 2020 using Point-of-care test and Line immunoassay test in Vojvodina, Serbia.**

| Point of sero-prevalence survey | Variable | Age group (years) | | | | | Overall (n) |
|---|---|---|---|---|---|---|---|
| | | 0–4 (n) | 5–14 (n) | 15–29 (n) | 30–64 (n)' | ≥65 (n) | |
| End of April | A. Cumulative number of COVID-19 [a] | 5 | 14 | 82 | 428 | 133 | 662 |
| | B. Cumulative number of ARI | 9,300 | 20,492 | 16,864 | 28,239 | 3,933 | 78,829 |
| | C. Estimated number of subjects with previous contact with SARS-CoV-2 | 0 | 3,190 | 6,431 | 33,035 | 9,908 | 50,227 |
| | Ratio B vs. A | 1860.0 | 1463.7 | 205.7 | 66.0 | 29.6 | 119.1 |
| | Ratio C vs. A | **NA** | **227.9** | **78.4** | **77.2** | **74.5** | **75.9** |
| End of May | A. Cumulative number of COVID-19 [a] | 10 | 24 | 120 | 551 | 170 | 875 |
| | B. Cumulative number of ARI | 11,767 | 23,298 | 19,874 | 32,910 | 5,077 | 92,926 |
| | C. Estimated number of subjects with previous contact with SARS-CoV-2 | 0 | 5,323 | 12,317 | 45,411 | 10,256 | 75,920 |
| | Ratio B vs. A | 1176.7 | 970.8 | 165.6 | 59.7 | 29.9 | 106.2 |
| | Ratio C vs. A | **NA** | **221.8** | **102.6** | **82.4** | **60.3** | **86.8** |
| End of June | A. Cumulative number of COVID-19 [a] | 14 | 32 | 246 | 872 | 217 | 1,381 |
| | B. Cumulative number of ARI | 14,547 | 26,487 | 23,453 | 38,601 | 6227 | 109,315 |
| | C. Estimated number of subjects with previous contact with SARS-CoV-2 | 0 | 5,643 | 23,107 | 67,531 | 17,694 | 118,033 |
| | Ratio B vs. A | 1039.1 | 827.7 | 95.3 | 44.3 | 28.7 | 79.2 |
| | Ratio C vs. A | **NA** | **176.3** | **93.9** | **77.4** | **81.5** | **85.5** |
| End of September | A. Cumulative number of COVID-19 [a] | 48 | 77 | 1,198 | 5,285 | 1,574 | 8,182 |
| | B. Cumulative number of ARI | 30,828 | 50,390 | 43,342 | 72,124 | 12,015 | 208,699 |
| | C. Estimated number of subjects with previous contact with SARS-CoV-2 | 9,174 | 24,518 | 60,893 | 157,281 | 29,691 | 282,044 |
| | D. Estimated number of subjects with previous contact with SARS-CoV-2 using line immunoassay test | 7,027 | 23,914 | 77,170 | 168,975 | 42,479 | 322,033 |
| | Ratio B vs. A | 642.3 | 654.4 | 36.2 | 13.6 | 7.6 | 25.5 |
| | Ratio C vs. A | **191.1** | **318.4** | **50.8** | **29.8** | **18.9** | **34.5** |
| | Ratio D vs. A | 146.4 | 310.6 | 64.4 | 32.0 | 27.0 | 39.4 |
| | **Ratio D vs. C** | **0.8** | **1.0** | **1.3** | **1.1** | **1.4** | **1.1** |

[a]Confirmed by the RT-qPCR;

NA-not applicable; ARI- acute respiratory infection.

Finally, the expected number of SARS-CoV-2 seropositive inhabitants of Vojvodina at the end of the study extrapolated from obtained SARS-CoV-2 seropositivity rates and estimated population size is 322,033, corresponding to an overall seroprevalence of 17%. The estimated total, minimum and maximum number of SARS-CoV-2 seropositive inhabitants of Vojvodina by age groups over study period are presented in S1 Table.

## Discussion

To the best of our knowledge, this is the first SARS-CoV-2 seroprevalence study in our country among asymptomatic subjects, and the first study that compared the results obtained from surveillance of ILI, ARI, official COVID-19 laboratory-confirmed cases and the seroprevalence of antibodies against SARS-CoV-2. The main findings from this seroprevalence study performed prior to the ongoing third epidemic wave of COVID-19 (end of September 2020) indicate that the prevalence of SARS-CoV-2 antibodies was around 17% with an estimated total number of 322,033 infections in Vojvodina. These results suggest that the population of Vojvodina has still not reached the desirable level of herd immunity with the majority of our population

being immunologically naive to SARS-CoV-2 virus, which explains the occurrence and the intensity of the ongoing third wave of the epidemic.

Seroprevalence of SARS-CoV-2 antibodies in general population varies across different regions and countries and ranges from < 0.1% to more than 20% [2–8, 17, 18]. Such large discrepancies could be ascribed to differences in the phases of epidemic, type of implemented measures of outbreak control, duration of the study period and type of used methodology, participant selection method, as well as serologic tests used. Common feature in these studies is low overall seroprevalence of SARS-CoV-2 antibodies in the population during the first and second waves of COVID-19 pandemic. Bearing this in mind, it is probable that population of most countries is still susceptible to SARS-CoV-2 infection. However, the results of anti-SARS-CoV-2 seroprevalence in other settings showed certain similarities with the findings on our territory, particularly in the early phase of COVID-19 pandemic. A population-based study in Switzerland [2] which was conducted during five consecutive weeks (between April 6 and May 9, 2020) reported seroprevalence between 4.8% (first week) and 10.8% (fifth week). A Spanish nationwide population-based seroepidemiological study [19] that was conducted from April 27 to May 11, 2020, demonstrated IgG antibodies against SARS-CoV-2 in 5% of Spanish population. At the same time, our survey revealed low prevalence of SARS-CoV-2 antibodies in our population during the first wave, since it was 2.6% at the end of April and 3.9% at the end of May. Lower seroprevalence of SARS-CoV-2 antibodies in our territory compared to the population in the two aforementioned studies is probably the result of the fact that Serbia was under a complete lockdown between March 16 and April 21, 2020 [9].

Interestingly, our study results indicate that there were around 76,000 citizens of Vojvodina with SARS-CoV-2 antibodies during the period of complete lockdown. Of note, after this lockdown, on May 22, 2020, Serbia completely opened the borders and relieved all restrictive measures introduced before [9]. As a result, a marked increase in the prevalence of official laboratory-confirmed COVID-19 cases and the seropositivity of anti-SARS-CoV-2 was detected. Our estimates are that at the end of June, there were around 118,000 and at the end of September as much as 322,000 citizens in Vojvodina, who had been in contact with SARS-CoV-2 virus, as assessed by the presence of anti-SARS-CoV-2 antibodies in our tested participants. The obvious increase in the number of seropositive people in late September, could be associated to a summer peak of the second wave of the epidemic that occurred between the second half of June and the first half of September.

Considering the fact that seroprevalence studies provide estimates of the total number of infections in a community (which is especially useful for recognition of portion of asymptomatic and mild cases who did not need to seek health care), we presume that over the course of our study in Vojvodina, there were 39–87 undetected infections for every RT-PCR confirmed case of COVID-19. Although the differences in the number of infections and officially reported cases in other regions might be even higher and range between 82 and 130 vs. 1 [17], it still appears that the coverage of testing in the population of Vojvodina was quite low. For example, in settings with a wide test-track-trace approach, it has been shown that the ratio between estimated seroprevalence SARS-CoV-2 infections and RT-PCR laboratory-confirmed decreased to 10 vs. 1 [20].

It is noteworthy to emphasize that the testing practices changed during the course of the pandemic in our territory. During the early phase of epidemic, the priority for testing was applied in symptomatic and severe clinical forms of COVID-19, and the testing of close contacts of laboratory-confirmed COVID-19 cases was not performed. Over time, with the improvement of RT-PCR diagnostic capacities and increased testing rates throughout Vojvodina, the indications for testing were expanded with permanent efforts to enhance testing among asymptomatic persons who had contact with laboratory-confirmed COVID-19 cases.

In addition, testing based on a personal request of citizens was also included. All these reasons probably contributed to a decrease in the ratio between officially registered and estimated cases by survey between April to September. Nevertheless, a large disproportion between SARS-CoV-2 infection and laboratory-confirmed COVID-19 cases strongly suggests that many of the patients have not been recognized and were underreported, so they could spread infection as asymptomatic mobile carriers in the population of Vojvodina. This disproportion was the most noticeable among children aged 5–14 years. A possible explanation for this may lie in the fact that COVID-19 among children commonly follows mild or asymptomatic clinical course and therefore cases remain underreported.

We have observed important disparities between different age groups of participants. During most of the investigation period, the highest seroprevalence of SARS-CoV-2 antibodies was detected in participants aged 30–64 years, but after completion of the study (the end of September), we have found the highest anti-SARS-CoV-2 seropositivity among subjects aged 15–29 years. This finding potentially indicates that the main reservoirs of SARS-CoV-2 infection in our population were students from high schools and faculties and young working people, probably due to their more frequent social interactions as well as a lower awareness of the need to follow recommended measures of control of the epidemic. On the other hand, the young children (0–4 years) and older people (≥65 years) had the lowest seroprevalence of all age groups. The post-infection immune response in children is not yet clear, but IgG response in children seems to be delayed [2]. As for the elderly, due to targeted efforts that were especially focussed to protect the oldest part of population in Vojvodina (through limitation of their close contacts), people in this age group were exposed to a significantly lower risk to the SARS-CoV-2 virus than the individuals belonging to other age groups. Since the immune response is impaired to some extent in the elderly [21], the possibility of false negative serology test results in this age group has to be taken into account when interpreting our results. Noteworthy the findings related to seroprevalence by age groups in our study are consistent with the results of previously published results from other regions across the world [2, 17, 19, 22, 23].

Apart from comparison of RT-PCR confirmed cases with the estimated seroprevalence cases, we have also showed that a tracking of ILI and ARI cases during this period of pandemic apparently coincided with the trend of registered COVID-19 cases. Although we are fully aware that not all registered ILI or ARI cases represented COVID-19 cases, still it is important to highlight that an increasing weekly number of ILI and ARI cases based on well-established active surveillance system of influenza disease can indirectly signalize the beginning of a new epidemic wave of COVID-19. This follow-up is especially important in late spring and during summer months when small portion of other respiratory viruses exist in the population.

Our study had some limitations. First, we did not test samples for virus neutralization and therefore, the presence and proportion of neutralizing IgG SARS-CoV-2 antibodies in our study group remained unknown. Second, we did not perform validation of used serological tests. Thus, considering the specificities of used tests, we cannot completely exclude a possibility that some of the participants had false positive results due to a past or present infection with other viruses, such as endemic non-SARS-CoV-2 coronavirus strains [24, 25]. However, we believe that our results are comparable with results presented in other studies, especially considering the fact that we have only included participants within a minimum three weeks after the peaks of epidemic or during an inter-epidemic period (endemic phase) of COVID-19 in Vojvodina when a diagnostic sensitivity of serological test (both IgM and IgG) has already reached a satisfactory level [6, 13, 15, 16]. Moreover, in the last round of the study, we have also used the test for detection of S1 subunit of the spike protein and N

protein of SARS-CoV-2 virus that improved diagnostic accuracy [26] regarding adequate classification of seropositive persons. The finding that, after the adjustment by more specific assay, a number of seropositive individuals increased and not decreased in most age groups (with the exception to some extent in children) argues against significant detection of the antibodies to endemic coronaviruses that has been misinterpreted as anti-SARS-COV-2 antibodies in our study participants. In any case, the use of rapid tests for population-based estimates, and particularly for monitoring trends over time, is in our opinion acceptable [27]. Third, our questionnaire did not predict collection of the data related to underlying conditions which might have biased the final seroprevalence findings. Fourth, due to a small sample size of participants aged 0–4 and 5–14 years, seroprevalence estimates presented wide confidence intervals in these age groups. Therefore, a further and more extensive study aimed to elucidate levels of seropositivity of SARS-CoV-2 in this population is warranted. Finally, the number of officially laboratory-confirmed COVID-19 cases in Vojvodina increased substantially since the finalization of the serosurvey, meaning that, the seroprevalence of SARS-CoV-2 antibodies reported in this study does not represent true seroprevalence at the time of publication.

In conclusion, results of our systematic sampling approach based on a 0.1% of the total population in Vojvodina have provided important seroprevalence data for national and local public health policies. Taking into account that the threshold of herd immunity for COVID-19 is estimated to be approximately 60%–70% [20, 28, 29], it is likely to assume that the next COVID-19 epidemic wave(s) is/are going to occur in the near future in our territory. On the other hand, considering the facts that a mere antibody detection without antibody function assessment does not equate protection and that a duration of seroprotection remains unclear to date [18, 30], the intensity of the next epidemic wave in our territory is unpredictable. Further study based on the measurement of the neutralizing antibody levels, which will better determine the immunity to reinfection in the population of Vojvodina, should be undertaken at the earliest opportunity. Until then, the steps aimed to help prevent the spread of SARS-CoV-2 virus in the community such as person to person distancing, consistent and correct use of face coverings, washing hands frequently as well as staying home when sick should be consistently applied [23, 30, 31]. Apart from these epidemiological measures, a mass immunization is crucial in controlling the COVID-19 epidemic [32, 33]. Vaccination campaigns, not only provide individual protection for those that have been immunized, but they also aim for herd immunity, given that a sufficient number of people got the vaccine [34]. By June 9, 2021, as much as 42% of population of Vojvodina has been vaccinated with at least one dose of vaccine [9], but the desirable threshold of 60–70% (or higher) of immune people has not been reached yet. In order to determine the true level of herd immunity for COVID-19 (both natural following infection and vaccine induced) in Vojvodina and Serbia, a prospective serosurvey parallel with rapid pandemic vaccination campaign, needs to be performed in the whole territory. The results provided through this future serosurvey, would be very important in targeted efforts to harmonize the public health policies that should lead eventually to mitigation and control of COVID-19 epidemic in our region.

## Supporting information

**S1 Table. Estimated number of SARS-CoV-2 seropositivity in population of Vojvodina, Serbia between April and September, 2020.**
(DOCX)

## Acknowledgments

The authors are grateful to all the nurses, general practitioners, paediatricians, administrative personnel, and other health care workers from Health Care Centres throughout Vojvodina who collaborated in this study and to all participantsthat voluntarily took part in the study. We also thank the epidemiologists from local departments of public health, namely dr Slađana Tomić, dr Olivera Stanišić, dr Tanja Todorović, dr Tatjana Medić, dr Gordana Cvetić, dr Marija Lazarević, dr Mina Jandrić Kočić, dr Tatjana Pecarski, dr Sandra Radlović, dr Dragica Kovačević, dr Nebojša Bohucki, dr Radivoj Filipov, dr Žanka Subić, and dr Svetlana Popov for their invaluable contributions to this study. A special thanks goes to Dr Jasmina Boban (Department for Radiology, Faculty of Medicine, University of Novi Sad, Novi Sad, Serbia) for the revision of the manuscript.

## Author Contributions

**Conceptualization:** Mioljub Ristić, Miloš Marković.

**Data curation:** Mioljub Ristić, Biljana Milosavljević, Slobodanka Vapa.

**Formal analysis:** Mioljub Ristić, Biljana Milosavljević, Slobodanka Vapa.

**Investigation:** Mioljub Ristić, Slobodanka Vapa.

**Methodology:** Mioljub Ristić, Biljana Milosavljević.

**Resources:** Mioljub Ristić, Slobodanka Vapa.

**Supervision:** Biljana Milosavljević.

**Validation:** Mioljub Ristić, Slobodanka Vapa.

**Visualization:** Mioljub Ristić, Slobodanka Vapa, Miloš Marković.

**Writing – original draft:** Mioljub Ristić, Miloš Marković, Vladimir Petrović.

**Writing – review & editing:** Miloš Marković, Vladimir Petrović.

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
