## [Decision Letter · Decision Letter 0]

4 Jun 2021

PONE-D-21-00866

Seroprevalence of antibodies against SARS-CoV-2 virus in Northern Serbia (Vojvodina): a four consecutive sentinel population-based survey study

PLOS ONE

Dear Dr. Ristić,

Thank you for submitting your manuscript to PLOS ONE. After careful consideration, we feel that it has merit but does not fully meet PLOS ONE’s publication criteria as it currently stands. Therefore, we invite you to submit a revised version of the manuscript that addresses the points raised during the review process.

I am sorry for the delay in making a decision on this paper. This was due to problems with finding  reviewers. The reviewer recommends that you make minor revisions to your manuscript. Please attend to all the concerns raised and then resubmit your revised manuscript as advised in this letter.

We look forward to receiving your revised manuscript.

Kind regards,

Martin Chtolongo Simuunza, PhD

Academic Editor

PLOS ONE

Journal Requirements:

Reviewers' comments:

Reviewer's Responses to Questions

**Comments to the Author**

1. Is the manuscript technically sound, and do the data support the conclusions?

Reviewer #1: Yes

2. Has the statistical analysis been performed appropriately and rigorously? 

Reviewer #1: Yes

3. Have the authors made all data underlying the findings in their manuscript fully available?

Reviewer #1: No

4. Is the manuscript presented in an intelligible fashion and written in standard English?

Reviewer #1: Yes

5. Review Comments to the Author

Reviewer #1: Thank you for the opportunity to review “Seroprevalence of antibodies against SARS-CoV-2 virus in Northern Serbia (Vojvodina): a four consecutive sentinel population-based survey study”.

The manuscript is well written, timely and has public health applications both locally and internationally. Please find below suggestions/comments that may be used to improve this manuscript.

1) In the introduction: a summary of the public health policies that were implemented in Vojvodina during the study period to provide context to the seroprevalence rates (i.e. lock downs, school closures, etc.)

2) In the methods:

a. There is a lot of detail in the methods on the immunoassays. This section should be more concise, and authors can refer to manufactures website or package inserts for more details.

b. Did the authors try to reweigh their sample to reflect the larger target population?

c. There is no mention in the methods of how seroprevalence rates were “adjusted” (as indicated table 1)?

d. Authors can try to use the Rogen Gladden equation to adjust for imperfect test characteristics when using one assay to determine seroprevalence, this is based on the sensitivity and specificity of the assay

3) Results:

a. Abbreviations (ARI) should be included as footnotes to each table

b. A summary of Table 2 would be easier to follow as graph

c. Did authors attempt to stratify by regions to identify “hot zones” within Vojvodina? There would presumably be significant heterogeneity by socioeconomic status.

d. Linear trend in figure 2 is not informative I would suggest removing, stratifying by age groups will illustrate results found in Table 2.

e. Is it possible to know if the study population reflect the same population that would be visiting the clinics (pre-covid), basic comparisons by demographic variables before and during covid would speak to the generalizability of the results to the target population

4) Conclusion:

a. Authors may want to comment on the feasibility of this study and steps to scale seroprevalence studies across the country or target specific areas.

b. Authors may also want to comment on vaccine rollouts in Vojvodina and how seroprevalence studies will continue to be important to track population level exposure.

6. PLOS authors have the option to publish the peer review history of their article (what does this mean?). If published, this will include your full peer review and any attached files.

Reviewer #1: No

---

## [Author Response · Author response to Decision Letter 0]

14 Jun 2021

PONE-D-21-00866

Seroprevalence of antibodies against SARS-CoV-2 virus in Northern Serbia (Vojvodina): a four consecutive sentinel population-based survey study

PLOS ONE

Dear Editor,

First of all, we would like to thank you for considering our paper for publication in your Journal. We are grateful for all the suggestions and remarks, which we find well-intended and helpful. The authors have considered all the comments and we did our best to correct the manuscript in the more appropriate and clear way. All the changes that have been made are highlighted in the manner proposed by the Journal. 

Response to Reviewers

IMPORTANT!

Comments to the Author are shown in regular fonts, while the comments to the academic Editor are in the italic fonts. 

Please note that in accordance with the Reviewer suggestions, we have revised the Figure 2 and attached the corrected file (Fig 2-corrected, in which we removed “linear trend” ). Also, the comment on vaccine rollouts in Vojvodina that was added in the Conclusion section, as suggested by the reviewer, resulted in addition of three new references in the revised version of the manuscript (references 32-34).

Reviewer #1: Thank you for the opportunity to review “Seroprevalence of antibodies against SARS-CoV-2 virus in Northern Serbia (Vojvodina): a four consecutive sentinel population-based survey study”.

The manuscript is well written, timely and has public health applications both locally and internationally. Please find below suggestions/comments that may be used to improve this manuscript.

1) In the introduction: a summary of the public health policies that were implemented in Vojvodina during the study period to provide context to the seroprevalence rates (i.e. lock downs, school closures,etc.)

-Thank you very much for this useful suggestion. In order to provide context to the seroprevalence rates, we added in the revised manuscript the short overview of the public health policies that were implemented in our territory during the study period.

2) In the methods:

a. There is a lot of detail in the methods on the immunoassays. This section should be more concise, and authors can refer to manufactures website or package inserts for more details.

-Thank you for this suggestion. However, considering a vast number of assays available and a lack of standardized methodology, the authors feel that it is very important to present all these laboratory procedures in detail in case that other readers/researches want to compare their results with ours. Other studies also explained laboratory procedures without citation of special reference, and many of them highlighted that procedures were performed according to manufacturer's instructions (Regarding this, we suggest to look for some of the references in the list provided below). Nevertheless, if the reviewer insists on shortentning of this section, we are ready to move this part to the supplementary material.

A list of references with similar presentation of laboratory procedures:

1. Chinese Center for Disease Control and Prevention. The Novel Coronavirus Pneumonia Emergency Response Epidemiology Team. The Epidemiological Characteristics of an Outbreak of 2019 Novel Coronavirus Diseases (COVID-19) — China, 2020. CCDC Weekly / Vol. 2.

2. Scohy A, Anantharajah A, Bodéus M, Kabamba-Mukadi B, Verroken A, Rodriguez-Villalobos H. Low performance of rapid antigen detection test as frontline testing for COVID-19 diagnosis. J Clin Virol. 2020;129:104455. doi: 10.1016/j.jcv.2020.104455. PMID: 32485618; PMCID: PMC7240272.

3. Corman VM, Landt O, Kaiser M, Molenkamp R, Meijer A, Chu DK, et al. Detection of 2019 novel coronavirus (2019-nCoV) by real-time RT-PCR. Euro Surveill. 2020;25(3):2000045. doi: 10.2807/1560-7917.ES.2020.25.3.2000045. PMID: 31992387; PMCID: PMC6988269.

4. World Health Organization. Diagnostic assessment: in vitro diagnostic medical devices (IVDs) used for the detection of high-risk human papillomavirus (HPV) genotypes in cervical cancer screening. Licence: CC BY-NC-SA 3.0 IGO. Available from: https://apps.who.int/iris/handle/10665/272282.

5. Assennato SM, Ritchie AV, Nadala C, Goel N, Tie C, Nadala LM, et al. Performance evaluation of the SAMBA II SARS-CoV-2 Test for point-of-care detection of SARS-CoV-2. J Clin Microbiol. 2020:JCM.01262-20. doi: 10.1128/JCM.01262-20. Epub ahead of print. PMID: 33051242.

6. Harrington A, Cox B, Snowdon J, Bakst J, Ley E, Grajales P, et al. Comparison of Abbott ID Now and Abbott m2000 Methods for the Detection of SARS-CoV-2 from Nasopharyngeal and Nasal Swabs from Symptomatic Patients. J Clin Microbiol. 2020;58(8):e00798-20. doi: 10.1128/JCM.00798-20. PMID: 32327448; PMCID: PMC7383519.

7. Mertens P, De Vos N, Martiny D, Jassoy C, Mirazimi A, Cuypers L, et al. Development and Potential Usefulness of the COVID-19 Ag Respi-Strip Diagnostic Assay in a Pandemic Context. Front Med (Lausanne). 2020;7:225. doi: 10.3389/fmed.2020.00225. PMID: 32574326; PMCID: PMC7227790.

8. Smithgall MC, Scherberkova I, Whittier S, Green DA. Comparison of Cepheid Xpert Xpress and Abbott ID Now to Roche cobas for the Rapid Detection of SARS-CoV-2. J Clin Virol. 2020;128:104428. doi: 10.1016/j.jcv.2020.104428. Epub 2020 May 13. PMID: 32434706; PMCID: PMC7217789.

9. Zhen W, Smith E, Manji R, Schron D, Berry GJ. Clinical Evaluation of Three Sample-to-Answer Platforms for Detection of SARS-CoV-2. J Clin Microbiol. 2020;58(8):e00783-20. doi: 10.1128/JCM.00783-20. PMID: 32332061; PMCID: PMC7383520.

10. Mak GC, Cheng PK, Lau SS, Wong KK, Lau CS, Lam ET, et al. Evaluation of rapid antigen test for detection of SARS-CoV-2 virus. J Clin Virol. 2020;129:104500. doi: 10.1016/j.jcv.2020.104500. Epub 2020 Jun 8. PMID: 32585619; PMCID: PMC7278630.

11. Caruana G, Croxatto A, Coste AT, Opota O, Lamoth F, Jaton K, et al. Diagnostic strategies for SARS-CoV-2 infection and interpretation of microbiological results. Clin Microbiol Infect. 2020;26(9):1178–82. doi: 10.1016/j.cmi.2020.06.019. PMID: 32593741; PMCID: PMC7315992.

12. Ristić M, Nikolić N, Čabarkapa V, Turkulov V, Petrović V. Validation of the STANDARD Q COVID-19 antigen test in Vojvodina, Serbia. PLoS One. 2021 Feb 22;16(2):e0247606. doi: 10.1371/journal.pone.0247606. PMID: 33617597; PMCID: PMC7899368.

b. Did the authors try to reweigh their sample to reflect the larger target population?

- Although we are aware that increasing the sample would better represent our population and improve our manuscript, we were not able to do it and we did not predict this in our serosurvey

c. There is no mention in the methods of how seroprevalence rates were “adjusted” (as indicated table 1)?

-Thank you for this helpful suggestion. In order to clarify that, we added in the “Data analysis” section additional information how the “adjustement” of the seroprevalence rates were done using results given by line immunoassay test.

d. Authors can try to use the Rogen Gladden equation to adjust for imperfect test characteristics when using one assay to determine seroprevalence, this is based on the sensitivity and specificity of the assay.

- Thank you for this interesting suggestion. We have tried to implement this equation, and obtained results, which were close to the values that we already had, especially after the adjustement of the results by line immunoassay test results (please refer to Table 1). Therefore we may assume that the final percentages that we got after the adjustment may well represent a true level of seroprevalence in Vojvodina at the moment when study was over (end of September, 2020).

3) Results:

a. Abbreviations (ARI) should be included as footnotes to each table.

- Thank you for this suggestion. We added abbreviations (ARI) in the Fig 2 and the Table 2 where the ARI was mentioned. We also changed a title of Table 2 accordingly.

b. A summary of Table 2 would be easier to follow as graph.

- Although we value this remark, we can not absolutely agree with it. Namely, Table 2 shows the whole spectrum of different numbers (cumulative, estimated), as well as ratios between cumulative and estimated number of COVID-19 and ARI by age groups during the four consecutive time-points in this seroprevalence survey. Thus, although the graph would be more easy to follow, it would be very complicated to show all of this data on one or more figures. Bearing this in mind, we think that Table 2 is more appropriate to present the full extent of our results in one place in the paper.

c. Did authors attempt to stratify by regions to identify “hot zones” within Vojvodina? There would presumably be significant heterogeneity by socioeconomic status.

- We did not manage to identify such “hot zones” within Vojvodina. As we stated in the Materials and methods section, Vojvodina is divided in 44 settlement areas and seven administrative districts. After the recruitment of the representative sample of participants (0.1% out of the total population) and assessment of the seroprevalence, we have not found that the prevalence of SARS-COV-2 seropositivity significantly deviated by districts or settlement areas of Vojvodina during the study period. It is possible that the reason for this lies in the small study sample. So, further and more extensive study in Vojvodina as well as in the whole territory of Serbia is warranted (as commented in Conclusion section in the revised manuscript).

d. Linear trend in figure 2 is not informative I would suggest removing, stratifying by age groups will illustrate results found in Table 2.

- We agree with your suggestion and we have removed “linear trend” in Fig 2. Corrected Figure has been provided together with the revised manuscript.

e. Is it possible to know if the study population reflect the same population that would be visiting the clinics (pre-covid), basic comparisons by demographic variables before and during covid would speak to the generalizability of the results to the target population.

- Thank you for this interesting question. We agree that basic comparisons by demographic variables before and during COVID would be very useful and speak to the generalizability of our results. However, as we stated in the Materials and methods section, we conducted a cross-sectional study from the representative sample of participants on a voluntary basis after controlling for age and gender, and we did not know, and this was not aim of this investigation, whether our participants visited the clinics or not. But, considering the inclusion criteria in this study, it seems that results of our study can be extrapolated to total population and whole territory of Vojvodina. 

4) Conclusion:

a. Authors may want to comment on the feasibility of this study and steps to scale seroprevalence studies across the country or target specific areas.

- In conclusion, we have already hinted that our results based on 0.1% of the population of Vojvodina may imply that similar results can be expected in the whole territory of Vojvodina, and Serbia as well. However, we are fully aware that “Further study based on the measurement of the neutralizing antibodies levels, which will better determine the immunity to reinfection in the population of Vojvodina, should be undertaken at the earliest opportunity” as we stated in the conclusion of this paper. We also accentuated the importance of scaling-up the seroprevalence studies across the whole country in the comment added in the Conclusion section in the revised manuscript.

b. Authors may also want to comment on vaccine rollouts in Vojvodina and how seroprevalence studies will continue to be important to track population level exposure.

- Thank you very much for this suggestion. We have added a comment on vaccine rollouts in Vojvodina in the conclusion section of the paper, and emphasized the importance of further seroprevalence studies for the estimation of herd immunity as a crucial step towards control of COVID-19 pandemics. As a result, three references has been added in the revised manuscript (References 32-34).

The authors feel that the paper is overall of a better quality after including the well-intended and helpful remarks obtained from the Reviewer. 

Thank you for the consideration of our manuscript, we look forward to the next step of the paper revisions and hope for the positive outcome. 

Sincerely, 

Mioljub Ristić, MD, PhD

Centre for Disease Control and Prevention, Institute of Public Health of Vojvodina, Novi Sad, Serbia

Futoška 121, Novi Sad 21 000, Serbia

E-mail: mioljub.ristic@mf.uns.ac.rs

---

## [Decision Letter · Decision Letter 1]

29 Jun 2021

Seroprevalence of antibodies against SARS-CoV-2 virus in Northern Serbia (Vojvodina): a four consecutive sentinel population-based survey study

PONE-D-21-00866R1

Dear Dr. Ristić,

We’re pleased to inform you that your manuscript has been judged scientifically suitable for publication and will be formally accepted for publication once it meets all outstanding technical requirements.

Kind regards,

Martin Chtolongo Simuunza, PhD

Academic Editor

PLOS ONE

Additional Editor Comments (optional):

Reviewers' comments:

Reviewer's Responses to Questions

**Comments to the Author**

1. If the authors have adequately addressed your comments raised in a previous round of review and you feel that this manuscript is now acceptable for publication, you may indicate that here to bypass the “Comments to the Author” section, enter your conflict of interest statement in the “Confidential to Editor” section, and submit your "Accept" recommendation.

Reviewer #1: All comments have been addressed

2. Is the manuscript technically sound, and do the data support the conclusions?

Reviewer #1: Yes

3. Has the statistical analysis been performed appropriately and rigorously? 

Reviewer #1: Yes

4. Have the authors made all data underlying the findings in their manuscript fully available?

Reviewer #1: No

5. Is the manuscript presented in an intelligible fashion and written in standard English?

Reviewer #1: Yes

6. Review Comments to the Author

Reviewer #1: Authors resisted a few suggestions but made efforts to address others. I have no other suggestions at this point.

7. PLOS authors have the option to publish the peer review history of their article (what does this mean?). If published, this will include your full peer review and any attached files.

Reviewer #1: No

---

## [Editor Report · Acceptance letter]

1 Jul 2021

PONE-D-21-00866R1 

Seroprevalence of antibodies against SARS-CoV-2 virus in Northern Serbia (Vojvodina): a four consecutive sentinel population-based survey study 

Dear Dr. Ristić:

I'm pleased to inform you that your manuscript has been deemed suitable for publication in PLOS ONE. Congratulations! Your manuscript is now with our production department. 

Kind regards, 

on behalf of

Dr. Martin Chtolongo Simuunza 

Academic Editor

PLOS ONE